# Behavioral Proxy Conditioning for Financial Stress Scenario Generation with a Pretrained Diffusion Model

**Elena Kuular** [1]   **Junsuk Choe** [1]

## Abstract

Controllable financial scenario generation is challenging because historical crises are rare and difficult to model directly. To address this, we adapt a pretrained diffusion model for financial time-series generation using interpretable behavioral proxy conditioning. The proxy combines HMM regime probabilities, cross-market correlation, and realized volatility rank, allowing the model to generate calm and stress scenarios for seven global equity indices. To make conditioning more robust, we randomly drop proxy groups during training, allowing the model to use information from all components. These design choices enable the model to generate realistic financial scenarios that remain responsive to regime conditions. The final model passes $20/21$ risk/tail checks plus $7/7$ volatility sanity checks, for a total of $27/28$ regime-separation checks across all four seeds with an average stress-to-calm volatility ratio of $(2.47 \pm 0.08)\times$ and $0\%$ exact replay of historical windows. Portfolio-level stress scenarios also show substantially worse $CVaR_{95}$ than calm scenarios, suggesting that behavioral proxy conditioning can make pretrained generative models effective for downstream financial stress testing under data scarcity.

## 1. Introduction

Interest in financial stress testing has increased following major crises such as the 2008 global financial crisis and the COVID-19 pandemic, when markets experienced extreme volatility and liquidity shortages that exposed weaknesses in risk-management frameworks. Today, financial institutions widely use stress testing to reduce uncertainty for portfolio construction, risk assessment, and capital planning (Cont, 2001). However, real financial crises are historically rare, creating a serious data-scarcity problem: abundant calm-market observations, but relatively few stress ones for model training.

This is why synthetic data generation is useful here. Still, since historical data is dominated by calm periods, a generator may reproduce normal market behavior while failing to reflect the stress regime. Generated stress scenarios therefore should not only look realistic, but also have economically meaningful characteristics, including higher volatility and more adverse tail risk than calm scenarios (Cont, 2001; Rockafellar & Uryasev, 2000).

Traditional approaches have important limitations. Parametric models such as GARCH are interpretable and useful for volatility modeling, but they are not designed to directly generate diverse scenarios under calm or stress regime conditions (Bollerslev, 1986). Since Historical Bootstrap draws on past data, it only resamples observed data and cannot generate genuinely new crisis scenarios (Künsch, 1989). More recent deep generative models, including TimeGAN (Yoon et al., 2019), QuantGAN (Wiese et al., 2020), and FinGAN (Takahashi et al., 2019) have shown good results, but they usually require substantial data and may struggle with calm and stressed regime transitions (Kwon & Lee, 2024).

Diffusion models have recently emerged as a strong alternative for financial data generation (Yuan & Qiao, 2024). Compared to recent conditional methods such as CoFinDiff (Tanaka et al., 2025) and InterDiff (Long et al., 2025), our model uses a compact and interpretable conditioning vector designed for stress testing. Specifically, we extend the pretrained ImagenFew (Gonen et al., 2025) backbone using a five-dimensional behavioral proxy including HMM regime probabilities, cross-market correlation, and window volatility rank (Hamilton, 1994). We also introduce proxy-group dropout to reduce reliance on any single conditioning signal and improve robustness. Consequently, the model can generate controllable calm and stress scenarios for each of seven global equity indices across North America, Europe, and Asia.

Our final Dropout-FT model, which combines proxy-group dropout and kurtosis-aware fine-tuning, achieves strong and

---

[1]Department of Computer Science and Engineering, Sogang University, Seoul, South Korea. Correspondence to: Elena Kuular <eakuular@sogang.ac.kr>.

*Proceedings of the $2^{nd}$ ICML Workshop on Foundation Models for Structured Data*, Seoul, South Korea. 2026. Copyright 2026 by the author(s).

consistent results across all four training seeds. It passes 27/28 checks in every run and achieves an average stress-to-calm volatility ratio of $(2.47 \pm 0.08)\times$, showing that the learned conditioning captures meaningful stress-regime behavior.

In summary, this work makes three contributions: (1) a compact behavioral proxy for controllable financial scenario generation, (2) a proxy-group dropout strategy for more robust conditioning, and (3) a seven-index stress-testing evaluation showing strong regime separation with zero historical replay.

## 2. Methodology

Our methodology uses behavioral proxy conditioning to improve controllable financial scenario generation. As summarized in Figure 1, the workflow consists of five blocks organized into three main stages: data preparation, proxy construction, and conditional generation. First, seven global stock indices are preprocessed. Second, we build a behavioral proxy from each window that represents latent regime state, market co-movement, and volatility level. Finally, the pretrained model is fine-tuned with a market identity token and the behavioral proxy so that it can generate controllable calm and stress scenarios.

### 2.1. Behavioral Proxy Construction

To enable explicit regime control, the generator is conditioned on a behavioral proxy that provides an interpretable and market-observable representation of the current market state. For each window $i$, the proxy is written as:

$$c_i = \left[ p_i^{\text{calm}}, p_i^{\text{normal}}, p_i^{\text{stress}}, \rho_i, v_i \right].$$

As shown in the proxy block of Figure 1, the first three components are soft regime probabilities from a three-state Gaussian HMM (Hamilton, 1994). The HMM is fitted on the training set only, using a rolling-volatility computed from close-channel log returns. We then sort the three HMM states in ascending order of average volatility and label them as "calm", "normal", and "stress". The fourth component, $\rho_i$, is the average cross-market correlation within the window. The fifth component, $v_i$, measures the realized volatility rank of the window. These three components give a generator a compact description of the window: its latent regime, market synchronization, and a volatility level.

To prevent the model from over-relying on one conditioning signal during training, we use stochastic proxy-group dropout with $p = 0.3$ and group weights $[1, 1, 2]$, where the volatility-rank group is dropped more often. It randomly masks one proxy group and encourages the model to use the remaining components.

Finally, the proxy vector is globally normalized using training-set statistics so that proxy magnitudes remain comparable across markets. A detailed justification for the proxy components and the normalization formula is provided in Section A.

### 2.2. Data Generation Pipeline

We use daily Open, High, Low, Close, Volume (OHLCV) data for seven stock indices: SPX, FTSE, GDAXI, N225, SENSEX, FCHI, and AEX, covering the period 2000-2024 and providing economic and geographical diversity. The data is split chronologically into train, validation, and test sets in a $70/15/15$ ratio. Prices are then converted into daily log returns and normalized using training-set statistics. After that, the time series is divided into 36-day sliding windows with a stride of 9. The first two blocks of Figure 1 correspond to the data and preprocessing steps.

A key preprocessing choice is whether to normalize each window by its own volatility using local volatility normalization (LVN). LVN can simplify optimization but may erase the volatility information that separates regimes. Therefore, the main Dropout-FT model uses LVN-off preprocessing, while the LVN-on version is kept as an ablation.

## 3. Experiments

Here, we describe the empirical setup to evaluate whether the generated data is realistic, financially meaningful, and controllable across calm and stress regimes. We assess the model using standard generative metrics together with stress-testing checks based on volatility, tail risk, and kurtosis.

### 3.1. Experimental Setup

**Base model.** Our base model is ImagenFew (Gonen et al., 2025), a diffusion model pretrained on 19 time-series datasets and specifically designed for data-scarce settings. We fine-tune it on financial data and condition it on both the market identity token and the behavioral proxy. Additional details are provided in Section B.

**Metrics.** We evaluate the model from two main perspectives: generation realism and conditioning quality. Generation realism is evaluated using discriminative score and context FID, where lower values are better (Yoon et al., 2019; Gonen et al., 2025). The discriminative score, $disc = |Acc. - 0.5|$, measures how easily a classifier can separate real and generated windows, while context FID (cFID) measures the distance between real and generated data distributions.

**Stress testing protocol.** For each index, we build calm and stress conditions using the bottom and top 20% of training windows ranked by per-index volatility rank. Volatility rank is used to define the evaluation split as it is deterministic, di-

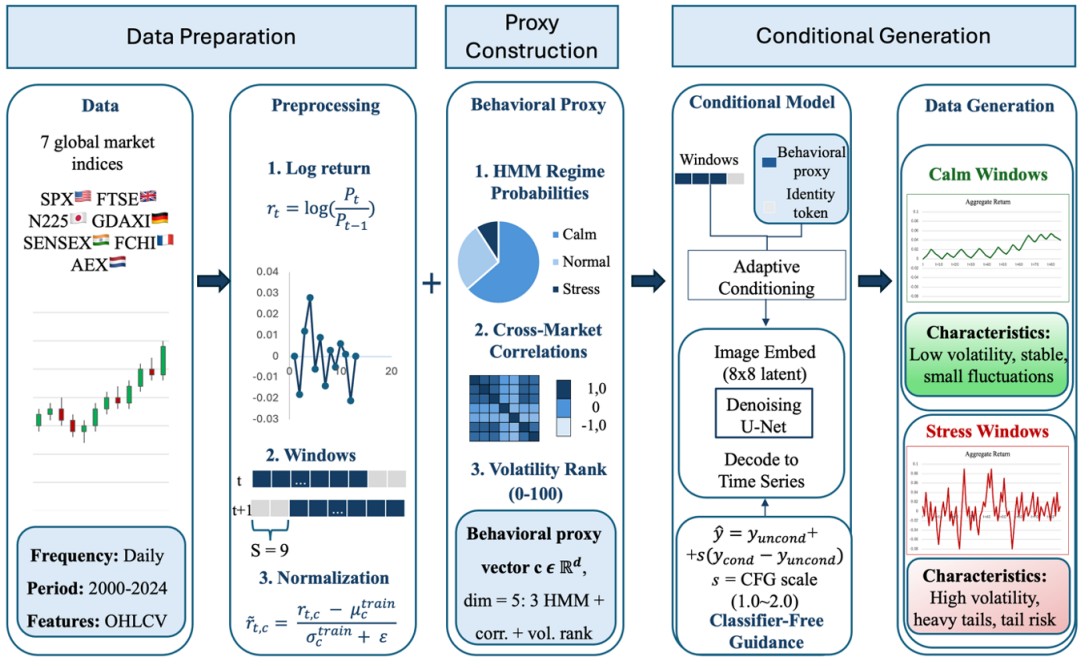

*Figure 1.* The pipeline has three stages: data preparation (1), proxy construction (2), and conditional generation (3). Daily OHLCV data from seven global market indices are transformed into log-return windows and normalized using training-set statistics only. A behavioral proxy is then computed from HMM regime probabilities, cross-market correlation, and realized volatility rank. The proxy and the index token guide the conditional ImagenFew diffusion model, which then generates synthetic calm and stress windows with different risk and tail properties.

rectly observable, and comparable across indices. Because volatility rank is part of the behavioral proxy, the volatility-ratio check also serves as a sanity check on whether the model follows the conditioning signal at all. The remaining three checks (CVaR$_{95}$, worst-step CVaR$_{99}$, and kurtosis direction) are stricter stress-testing criteria. We therefore treat the seven volatility checks as *controllability* evidence (vol. rank defines both the proxy and the split) and the risk/tail checks, together with the held-out-real portfolio of Section D, as the *independent* evidence (detailed in Section 3.2). The generator itself is still conditioned on the full five-dimensional behavioral proxy. We then generate 500 synthetic samples per regime and test whether the stress scenarios produce higher realized volatility (1), more adverse Conditional Value-at-Risk at the 95% confidence level (CVaR$_{95}$) (2), more adverse worst-step CVaR$_{99}$ (3), and the expected kurtosis direction (4) (Cont, 2001; Rockafellar & Uryasev, 2000). It produces 7 indices × 4 checks = 28 regime-separation checks in total, consisting of 7 volatility sanity checks and 21 risk/tail checks (check definitions are in Section A).

**Implementation details.** The third block of Figure 1 shows how the market identity token and behavioral proxy are passed to the ImagenFew backbone. We further improve controllability with classifier-free guidance (CFG) (Ho &

Salimans, 2022). During training, the full proxy is removed in a subset of batches (10%) so that one model learns both unconditional and conditional behavior. During inference, CFG combines the two predictions through a guidance strength $s$: $\hat{y}_{\text{cfg}} = \hat{y}_{\text{uncond}} + s(\hat{y}_{\text{cond}} - \hat{y}_{\text{uncond}})$, where $\hat{y}_{\text{cond}}$ and $\hat{y}_{\text{uncond}}$ are the conditional and unconditional model predictions at the same diffusion step. We select $s = 1.6$ on the validation split, which gives the best empirical balance between realism and stress/calm separation (Section B.7). Optimizer, learning-rate schedule, epoch counts, batch size, and hardware are summarized in Table B.1.

### 3.2. Main results

The stress signal of the final Dropout-FT model remains stable across seeds. As summarized in Table 1, it passes all 7/7 volatility sanity and 20/21 risk/tail checks, giving 27/28 checks across all four training seeds, with an average stress-to-calm volatility ratio of $(2.47 \pm 0.08)\times$. Among the 21 checks, 14 are CVaR-based tail-risk checks that do not use the axis defining the split; they test a different financial property — tail loss rather than dispersion — and provide evidence that is not mechanically tied to the conditioning signal, and they pass 14/14. Regime-aligned portfolio aggregation gives the same stress-testing signal: in normalized close-channel space, stress scenarios have sub-

*Table 1.* Performance of the proposed Dropout-FT model across all four training seeds. Portfolio $CVaR_{95}$ values are in normalized space (close channel after the training z-score, summed over the 36-day window)

| Metric | Value |
|---|---|
| Volatility sanity checks passed | 7/7 |
| Risk/tail checks passed | 20/21 |
| Total stress-testing checks passed | 27/28 |
| Average stress/calm volatility ratio | $(2.47 \pm 0.08)\times$ |
| Stress $CVaR_{95}$ (portfolio) | $-4.33 \pm 0.25$ |
| Calm $CVaR_{95}$ (portfolio) | $0.77 \pm 0.15$ |
| Discriminative score | $0.223 \pm 0.003$ |
| cFID | $2.069 \pm 0.076$ |
| Exact replay | $0\%$ |

*Table 2.* Baseline comparison under the same stress-testing protocol

| Method | Checks | Vol. ratio | Exact replay | Control |
|---|---|---|---|---|
| GBM | 24.8/28 | $3.69\times$ | 0% | Weak |
| Historical Bootstrap | 27.3/28 | $3.55\times$ | 100% | Limited |
| FinGAN | 22.0/28 | $3.25\times$ | 0% | Partial |
| Our model | 27.0/28 | $2.47\times$ | 0% | Yes |

stantially worse $CVaR_{95}$ than calm scenarios, with averages of $-4.33 \pm 0.25$ and $0.77 \pm 0.15$, respectively. The calm value is non-negative because calm 36-day windows have mild positive drift in normalized space.

The model also maintains competitive realism, with an average discriminative score of $0.223 \pm 0.003$ and average cFID of $2.069 \pm 0.076$. We further evaluate stylized facts beyond kurtosis, including return autocorrelation, volatility clustering, the leverage effect, tail heaviness, and OHLCV cross-channel consistency (Section E): the model robustly reproduces cross-channel structure, the leverage effect, and non-Gaussianity, with volatility clustering largely preserved. Converting the generated scenarios to approximate return units preserves the stress/calm separation (a stress-minus-calm 36-day CVaR95 gap of about -7%). Bootstrap resampling also gives 27.0/28 with tight volatility-ratio intervals across seeds. Additional per-index metrics and bootstrap results are in Section B; economic interpretation and distributional plots are provided in Section C.1.

The per-index results are consistent across markets. The weakest volatility ratio for Nikkei N225 is still $2.13\times$, and the strongest (SPX) reaches $2.80\times$. The only failed check is the GDAXI kurtosis-order check, but it still passes the volatility, $CVaR_{95}$, and worst-step $CVaR_{99}$ checks. Full per-index metrics are reported in Table B.6.

We further compare the final model with three external baselines: Geometric Brownian Motion (GBM), Historical Bootstrap (Künsch, 1989), and an adapted FinGAN baseline (Takahashi et al., 2019). As shown in Table 2, they represent different trade-offs between novelty and controllability.

Historical Bootstrap passes all checks, but the result should be interpreted carefully because the method also has a 100% exact replay rate. Therefore, its performance mainly reflects historical reuse rather than true scenario generation. GBM and FinGAN, in contrast, produce new samples, but their stress control is weaker. Among the non-replay methods evaluated under the same protocol, our model gives the

strongest stress-separation result, achieving 27/28 checks with 0% exact replay. Nearest-neighbor distance (Table B.2) and Figure 2 further support this.

### 3.3. Downstream portfolio (held-out real returns)

Beyond sample-level checks, we feed the generated calm/stress scenarios into a long-only stress-aware optimizer and evaluate the resulting weights on *held-out real* returns (Section D). The portfolio built from our scenarios attains the best downside-risk point estimates among all methods—lowest maximum drawdown ($-14.98\%$), highest Calmar (0.60), and least adverse 36-day $CVaR_{95}$ ($-7.50\%$). We report point estimates; a stationary-bootstrap check (Section D) indicates the downside-risk advantage is directionally stable across resamples, though confidence intervals are wide.

### 3.4. Ablation study

Ablations in Section B show that both proxy conditioning and LVN-off preprocessing are important for controllability.

The model's performance without the behavioral proxy drops from 27/28 to about 18.75/28 checks on average, while using LVN-on preprocessing reduces it to $(17.0 \pm 2.58)/28$.

## 4. Conclusion

We showed that a pretrained time-series diffusion model can be adapted to controllable financial stress scenario generation using a compact, interpretable behavioral proxy, without modifying the model architecture or relying on large domain-specific datasets. Across seven global equity indices and four training seeds, the proxy consistently separates calm and stress regimes, achieving 27 of 28 regime checks with no exact replay. The generated scenarios also preserve meaningful downside-risk differences in a held-out portfolio, suggesting that behavioral proxy conditioning is an effective way to adapt pretrained generative models to financial stress testing under data scarcity.

**Future Work.** Future work will focus on learning joint cross-asset dynamics and extending the framework to multivariate portfolio stress testing.

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

# A. Methodological Details and Design Justification

## A.1. Proxy Construction Details

**Local volatility normalization (LVN).** Local Volatility Normalization transforms each window $w$ as:

$$\hat{r}_{t,c}^{(w)} = \frac{\tilde{r}_{t,c}^{(w)}}{\mathrm{std}\left(\tilde{r}_{:,\,close}^{(w)}\right) + \varepsilon}$$

LVN preserves the internal shape of a window while removing scale effects, which may simplify optimization. However, it may also damage regime conditioning as it removes part of the volatility information that separates calm and stressed market states. Because every window is rescaled, the larger absolute moves and higher volatility of the stress window may also disappear, making regimes look more similar after normalization.

LVN may therefore improve some realism metrics, but it can reduce regime controllability.

**Proxy Vector Normalization Formula**

$$\tilde{c}_{i,d} = \frac{c_{i,d} - \mu_d^{global\ train}}{\sigma_d^{global\ train} + \varepsilon}.$$

## A.2. Justification of Design Choices

**Behavioral proxy components.** We design the behavioral proxy to be compact and interpretable. The goal is to give the generator a small set of signals that is useful for stress testing. We use three types of such signals.

First, we fit *a three-state Gaussian HMM* to obtain soft regime probabilities for the calm, normal, and stress regimes. The window may be partly calm and partly stressed, so soft probabilities allow the model to see the gradual change, giving a more realistic and flexible conditioning signal.

Second, *cross-market correlation*, represented by pairwise Pearson correlation across indices within a window, captures whether the shocks are local or systemic: For a window $i$ with $N = 7$ indices, define:

$$\rho_i = \frac{2}{N(N-1)} \sum_{1 \le a < b \le N} \mathrm{corr}\left(r_a^{(i)}, r_b^{(i)}\right),$$

where $r_a^{(i)}$ and $r_b^{(i)}$ are the close-channel log-return vectors for indices $a$ and $b$ inside window $i$. It typically represents a stressed window if many indices move together, and correlation is high. Low correlation, in contrast, tells that the shock is more local or index-specific. This component is especially important for portfolio-level stress testing.

Finally, *realized volatility rank* tells the model how extreme the window's volatility is. We do not use raw volatility as absolute volatility levels differ across markets. Rather, a 95th-percentile window means that it is unusually volatile for its own index, regardless of the index's absolute volatility scale, providing a model-free severity measure.

Overall, the proxy components encode regime state, the level of market synchronization, and stress magnitude. This gives the model more precise conditioning information while being easy to interpret.

**Stress-testing checks.** Stress-test metrics are used to check whether the generator separated calm and stress regimes in a financially meaningful way. In other words, a model may generate realistic financial windows but still fail in stress testing if calm and stress regimes produce similar samples. We use four checks for evaluation.

The *realized volatility* tests whether stress-conditioned scenarios have higher realized volatility than calm-conditioned ones. It is one of the most basic properties, capturing the overall size of market movements. We calculate realized volatility as the standard deviation of close-channel returns within each window.

*The total Conditional Value-at-Risk at the 95% confidence level* (CVaR$_{95}$) measures the expected loss in the worst 5% of outcomes over the window. It is a standard tail-risk metric in modern financial risk management. For stress scenarios, CVaR$_{95}$ should be more negative than for calm ones, showing more severe losses.

*Worst-step CVaR$_{99}$* captures sudden crash-like single movements, that may be hidden in total window statistics. It complements total CVaR$_{95}$ because some important events may be short, sudden, and sharp. Worst-step CVaR$_{99}$ tests if the generator produces realistic crash dynamics at the day-level, not just total outcomes.

*Excess kurtosis* measures how heavy-tailed a distribution is relative to a Gaussian (zero kurtosis) and tests whether the generated stress/calm ordering matches the empirical ordering in the real training windows.

Thus, these checks capture complementary properties of stress samples: higher volatility, worse tail losses, sharper crashes, and appropriate tail shape. Across all seven indices, it gives $7 \times 4 = 28$ regime-separation checks.

## B. Training Setup and Robustness

### B.1. Training Setup.

Table B.1 summarizes the main data, training, and generation settings used in the final experiments. These settings describe the LVN-off configuration and the later kurtosis fine-tuning step, which improves stress-tail behavior.

*Table B.1.* Training setup

| Setting | Value |
| --- | --- |
| Indices | S&P 500, FTSE 100, Nikkei 225, DAX, BSE SENSEX, CAC 40, AEX |
| Features | OHLCV |
| Period | 2000-2024 |
| Representation | Daily log-returns |
| Sequence length | 36 |
| Diffusion steps | 36 |
| Stride | 9 |
| Train/validation/test split | 70/15/15 |
| Normalization | Global z-score using training statistics |
| HMM rolling-volatility window | 5 trading days |
| Local volatility normalization | Off for main model; on for ablation |
| Behavioral proxy dimension | 5 |
| Samples per regime | 500 per index and regime |
| CFG scale | 1.0, 1.2, 1.4, 1.6, 1.8, 2.0, final $s = 1.6$ |
| Optimizer | AdamW |
| Learning rate | $2 \times 10^{-4}$ for base training; $2 \times 10^{-5}$ for fine-tuning |
| Weight decay | $1 \times 10^{-4}$ |
| Batch size | 512 |
| Proxy-group dropout | 0.3 |
| Training epochs | 500 for base model; 150 for kurtosis-aware fine-tuning |
| Kurtosis-aware fine-tuning | Applied after LVN-off base training |

### B.2. Baselines

CoFinDiff and InterDiff are discussed as related conditional diffusion work, but we do not include them as direct baselines because we could not identify official implementations for our setting. We use GBM, Historical Bootstrap, and FinGAN as practical baselines because they cover three different settings: parametric simulation, replay-based resampling, and GAN-based generation.

GBM tests whether a simple parametric simulator can reproduce regime-level volatility differences. Historical Bootstrap tests the strongest replay-based alternative: it uses real calm and stress windows, so it can pass many stress checks, but it cannot generate genuinely new scenarios. FinGAN tests a deep generative financial baseline under the same window length and preprocessing.

All baselines are evaluated with the same 28-check protocol, allowing us to compare stress separation, novelty, and controllability.

**Exact replay.** Exact replay is measured by comparing each generated OHLCV return window with training set windows in the normalized space. A replay is counted only when the full generated window matches a training window. The final

*Table B.2.* Nearest-neighbor distance check. The train-to-train leave-one-out (LOO) baseline measures the natural spacing among real training windows.

| Method | Mean 1-NN RMSE | Median 1-NN RMSE | Exact Replay | Interpretation |
|---|---|---|---|---|
| Train-to-train LOO | 0.929 | 0.820 | 0% | Natural real-data spacing reference |
| GBM | $1.141 \pm 0.001$ | $1.132 \pm 0.001$ | 0% | Least close to real data |
| Zero-cond. | $0.954 \pm 0.060$ | $0.952 \pm 0.070$ | 0% | Better than GBM, but worse than the conditioned model |
| Historical Bootstrap | $0.000 \pm 0.000$ | $0.000 \pm 0.000$ | 100% | Exact replay |
| FinGAN | $0.977 \pm 0.002$ | $0.948 \pm 0.002$ | 0% | Worse than zero-cond. and our model |
| Our model | $0.828 \pm 0.006$ | $0.817 \pm 0.004$ | 0% | Closest to real data while avoiding replay |

*Table B.3.* Ablation of proxy-group dropout and kurtosis-aware fine-tuning. Seed-0 rows isolate the effect of each training component, while the final row reports the four-seed result used in the main paper.

| Model | Dropout | Kurt. FT | Kurt. dir. | Checks | Vol. ratio |
|---|---|---|---|---|---|
| No-dropout base | No | No | – | $(23.3 \pm 0.5)/28$ | $(2.64 \pm 0.07)\times$ |
| Dropout base | Yes | No | – | $(23.8 \pm 0.5)/28$ | $(2.68 \pm 0.06)\times$ |
| Dropout + kurt. FT | Yes | Yes | window-level | $(25.8 \pm 0.5)/28$ | $(2.18 \pm 0.04)\times$ |
| Our model | Yes | Yes | pooled | $(27.0 \pm 0.0)/28$ | $(2.47 \pm 0.08)\times$ |

Dropout-FT model has $0\%$ exact replay.

Beyond exact replay, we also evaluate nearest-neighbor novelty. Our generated windows have a mean 1-NN RMSE of $0.828$ to the training set in comparison with the train-to-train baseline of $0.929$, while GBM and the no-proxy ablation are farther away; see Table B.2.

### B.3. Proxy-group Dropout and Kurtosis-Aware Fine-Tuning.

The base generator is trained with EDM/Karras diffusion objective (Gonen et al., 2025). In the final model, we apply an additional kurtosis fine-tuning to improve stress-tail behavior (Cont, 2001). In each batch, samples are split into stress-like ($v_i > 0.5$) and calm-like ($v_i < -0.5$) groups using the normalized volatility-rank proxy. The auxiliary loss penalizes cases where generated stress samples have lower excess kurtosis than generated calm samples:

$$\mathcal{L}_{\mathrm{kurt}} = \lambda_{\mathrm{kurt}} \max \left(0, \mathrm{kurt}_c - \mathrm{kurt}_s + m\right),$$

where $\mathrm{kurt}_s$ and $\mathrm{kurt}_c$ are the generated excess kurtosis values for stress-like and calm-like samples in the batch, $m = 0.1$ is the margin, and $\lambda_{\mathrm{kurt}}$ is the fine-tuning weight.

**Proxy-Group Dropout.** During training, we apply stochastic proxy-group dropout with probability $p = 0.3$ to prevent the model from relying too much on a single proxy component. In other words, exactly one of the three proxy groups is randomly masked for the selected sample, using proxy-group weights $[1, 1, 2]$ for HMM regime probabilities, cross-correlation, and volatility rank, respectively. The volatility-rank group is dropped more frequently because it is the strongest signal of stress market. This helps the model use the remaining proxy components instead of relying only on volatility rank.

Table B.3 isolates the contribution of each training component. The last two rows have the same dropout and kurtosis-aware fine-tuning setup but differ in how kurtosis-direction signs are estimated: window-level is less stable, and the pooled regime-level estimator used in our model gives a cleaner tail-shape correction, and the model reaches 27/28 in every seed.

### B.4. Ablation Details.

Table B.4 evaluates the effect of removing the behavioral proxy and changing the preprocessing from LVN-off to LVN-on. Removing the proxy reduces the average check count to $18.75 \pm 1.26/28$, and the average stress/calm volatility ratio decreases to $(1.01 \pm 0.01)\times$. This shows that the behavioral proxy carries meaningful information for stress/calm separation.

The LVN-on ablation also weakens controllability. Across four seeds, the average check count drops to $(17.0 \pm 2.58)/28$,

*Table B.4.* Ablation results across four training seeds.

| Method | Main change | Checks | Vol. ratio |
|---|---|---|---|
| Zero-cond. | No proxy | $18.75 \pm 1.26/28$ | $(1.01 \pm 0.01)\times$ |
| LVN-on | Per-window scale norm. | $17.0 \pm 2.58/28$ | $(0.94 \pm 0.02)\times$ |
| Our model | LVN-off | $27 \pm 0.0/28$ | $(2.47 \pm 0.08)\times$ |

*Table B.5.* Training-side realism metrics across four training seeds. Lower values are better.

| Model | LVN | Best disc. ↓ | Best cFID ↓ |
|---|---|---|---|
| LVN-on ablation | On | $0.160 \pm 0.005$ | $6.481 \pm 0.213$ |
| Our model | Off | $0.223 \pm 0.003$ | $2.069 \pm 0.076$ |

and the stress/calm volatility ratio decreases to $(0.94 \pm 0.02)\times$. This supports the main preprocessing choice: per-window scale normalization can remove the volatility magnitude that separates calm and stressed regimes. In contrast, the final Dropout-FT model, which uses LVN-off preprocessing, maintains the stress signal and achieves $27/28$ checks with an average stress-to-calm volatility ratio of $(2.47 \pm 0.08)\times$.

Table B.5 compares realism metrics of the final Dropout-FT LVN-off model with the LVN-on ablation. LVN-on may improve realism metrics after scale normalization, but it weakens regime controllability. For this reason, the final model is chosen not only for low discriminative score and cFID, but for the best overall balance between realism and stress controllability.

### B.5. Per-index Stress-Testing Results.

Table B.6 provides the full per-index stress-testing metrics supporting the overall $27/28$ regime-separation result.

For GDAXI, the empirical training-window split gives a small reversed kurtosis ordering, with excess kurtosis $1.245$ for calm windows and $1.177$ for stress windows. Since the margin is small, this index is the most fragile kurtosis-direction check in the benchmark.

### B.6. Bootstrap Confidence Interval (CI)

Table B.7 separates two types of stability. First, the model demonstrates consistent behavior across four random seeds. Each seed passes $27/28$ stress-regime checks with a bootstrap confidence interval of $[27.0, 27.0]$. In addition, the average stress-to-calm volatility ratio remains stable at approximately $2.5$, indicating stable stress/calm separation. These results support the claim that the model's controllability is robust rather than seed-dependent.

### B.7. Validation CFG Sweep

We select the CFG scale on the validation split before test evaluation. All scales pass $27/28$ checks, but larger guidance increasingly amplifies volatility and tail shape. We choose $s = 1.6$ as the best trade-off: it gives strong stress/calm separation while avoiding the stronger drift effects at $s = 2.0$.

*Table B.6.* Per-index stress-testing result for the final Dropout-FT model (seed 0). Subscripts $s$ and $c$ denote the stress and calm regimes, respectively. Volatility Ratio is computed as $\text{Vol}_s / \text{Vol}_c$, where Vol is the within-window realized volatility (close-channel standard deviation); the ratio is invariant under monotonic rescaling. $\text{Kurtosis}_s$ and $\text{Kurtosis}_c$ are excess kurtosis values for the stress and calm regimes.

| Index | $\text{Vol}_s$ | $\text{Vol}_c$ | Vol. Ratio | $\text{CVaR95}_s$ | $\text{CVaR95}_c$ | $\text{Kurt}_s$ | $\text{Kurt}_c$ | Real-train kurt. dir. |
|---|---|---|---|---|---|---|---|---|
| SPX | 1.315 | 0.469 | 2.80× | -16.62 | -2.63 | 2.16 | -0.11 | calm < stress |
| FTSE | 0.916 | 0.421 | 2.18× | -10.66 | -2.23 | 1.41 | -0.19 | calm < stress |
| GDAXI | 0.921 | 0.356 | 2.58× | -11.68 | -0.75 | 0.70 | -0.11 | calm > stress |
| N225 | 1.049 | 0.492 | 2.13× | -11.32 | -2.44 | 0.90 | -0.01 | calm < stress |
| SENSEX | 0.872 | 0.361 | 2.42× | -8.46 | -2.13 | 0.73 | -0.02 | calm < stress |
| FCHI | 0.929 | 0.413 | 2.25× | -10.99 | -1.33 | 1.09 | -0.14 | calm < stress |
| AEX | 0.919 | 0.402 | 2.29× | -9.01 | -1.47 | 2.05 | -0.18 | calm < stress |

*Table B.7.* Multi-seed robustness for our Dropout-FT model. Bootstrap confidence intervals are computed by resampling generated stress/calm windows for each fixed checkpoint.

| Seed | Checks | Avg. Vol. Ratio | Vol. Ratio 95% CI |
|---|---|---|---|
| s0 | 27/28 | 2.38× | [2.37, 2.40] |
| s1 | 27/28 | 2.57× | [2.55, 2.58] |
| s2 | 27/28 | 2.46× | [2.44, 2.47] |
| s3 | 27/28 | 2.49× | [2.48, 2.50] |
| Mean ± sd | 27.0 ± 0.0 | (2.47 ± 0.08)× | – |

*Table B.8.* Paired permutation test, full model vs. matched zero-proxy control, across the 28 stress checks. Positive means full > zero.

| Metric | $n$ pairs | Mean (full − zero) | Perm. $p$ |
|---|---|---|---|
| log volatility ratio | 28 | 0.895 | $< 10^{-4}$ |
| $\text{CVaR}_{95}$ separation | 28 | 7.42 | $< 10^{-4}$ |
| worst-step $\text{CVaR}_{99}$ sep. | 28 | 3.39 | $< 10^{-4}$ |

*Table B.9.* Validation sweep for CFG scale on the final Dropout-FT model. CVaR gaps are stress minus calm, so more negative values indicate stronger stress separation. Volatility ratios in this sweep are on the validation split used for CFG selection; the test-split ratio at s = 1.6 is 2.47× (Table 1).

| CFG | Checks | Vol. ratio | Mean total CVaR gap | Disc.↓ | cFID↓ |
|---|---|---|---|---|---|
| 1.0 | 27/28 | (2.53 ± 0.08)× | −10.58 ± 0.93 | 0.180 | 1.988 |
| 1.2 | 27/28 | (2.75 ± 0.09)× | −11.03 ± 1.09 | 0.168 | 2.056 |
| 1.4 | 27/28 | (2.94 ± 0.11)× | −11.42 ± 1.26 | 0.168 | 2.167 |
| 1.6 | 27/28 | (3.11 ± 0.13)× | −11.77 ± 1.43 | 0.166 | 2.250 |
| 1.8 | 27/28 | (3.28 ± 0.17)× | −12.10 ± 1.59 | 0.169 | 2.324 |
| 2.0 | 27/28 | (3.43 ± 0.20)× | −12.40 ± 1.74 | 0.171 | 2.392 |

# C. Additional Economic and Distributional Analysis

## C.1. Economic Interpretation

For economic interpretability, we approximately recover raw-return units by converting the normalized close-returns using the inverse train z-score. Because the final model is LVN-off, the recovered values preserve the original volatility scale better than with LVN-on.

We then build an equal-weight portfolio over the seven indices and evaluate 36-day CVaR95 together with worst-day CVaR99 for both stress and calm regimes.

The volatility ratios in Table 1 are computed in normalized close-channel signal space, while Table C.1 uses ratios after approximate inverse z-score conversion. Small differences are expected.

*Table C.1.* Economic-scale portfolio diagnostics after approximate raw-return conversion. Ratios may differ slightly from the normalized-space ratios in the main text.

| Metric | Normalized signal space | Approx. simple-return units | Interpretation |
|---|---|---|---|
| Stress/calm vol. ratio | $2.38\times$ | $2.33\times$ | Stress windows are about $2.3\times$ more volatile. |
| Stress 36-day $CVaR_{95}$ | $-4.480$ | $-5.37\%$ | Stress has much worse 36-day downside risk. |
| Calm 36-day $CVaR_{95}$ | $0.894$ | $1.65\%$ | Calm scenarios remain mild in this portfolio diagnostic. |
| Stress worst-day $CVaR_{99}$ | $-1.321$ | $-1.77\%$ | One-day tail losses become larger under stress. |
| Calm worst-day $CVaR_{99}$ | $-0.471$ | $-0.64\%$ | Calm one-day losses are much smaller. |
| Stress–calm 36-day $CVaR_{95}$ gap | $-5.373$ | $-7.02\%$ | Stress conditioning creates clear portfolio-level separation. |
| Stress–calm worst-day $CVaR_{99}$ gap | $-0.849$ | $-1.13\%$ | Stress also worsens short-horizon downside risk. |

The results show that all methods generate more adverse portfolio outcomes under stress than under calm conditions. For the final Dropout-FT model, the equal-weight seven-index portfolio has a 36-day $CVaR_{95}$ of $-5.37\%$ under stress, in comparison with $1.65\%$ under calm conditions. Overall, the learned stress regime remains meaningful not only in normalized space, but also after approximate conversion to raw-return units and portfolio aggregation.

## C.2. Distributional Comparison

Figure 2 shows the aggregate stress/calm separation of the final Dropout-FT model.

*Figure 2.* Aggregate stress/calm separation for our Dropout-FT model. The histogram and empirical CDF show that stress scenarios are more dispersed and more left-tailed than calm scenarios, supporting the regime-separation checks.

# D. Downstream Portfolio Stress Testing

We use the generated calm/stress scenarios as inputs to a long-only, stress-aware allocation problem: the optimizer minimizes downside risk under the stress scenarios subject to a calm-regime return floor. Because the generator produces assets

independently, we apply a regime-specific Iman–Conover rank-reordering (Iman & Conover, 1982) so stress (resp. calm) scenarios match the cross-asset correlation of real stress (resp. calm) periods, then blend the optimizer weights with equal-weight 80/20 to limit instability. All four seeds are pooled; weights are evaluated on *held-out real* test returns, gross of costs, with a 2% risk-free rate.

*Table D.1.* Downstream portfolio comparison on held-out real returns. Calmar, MDD, and $\text{CVaR}_{95}$ are the stress-aligned metrics; Sharpe/Sortino/return are secondary.

| Method | Ann. Ret. | Sharpe | Sortino | Calmar | MDD | Daily $\text{CVaR}_{95}$ | 36d $\text{CVaR}_{95}$ |
|---|---|---|---|---|---|---|---|
| GBM | 8.97% | 0.63 | 0.83 | 0.54 | $-16.58\%$ | $-1.75\%$ | $-7.98\%$ |
| Hist. Bootstrap | 8.95% | 0.63 | 0.83 | 0.54 | $-16.60\%$ | $-1.75\%$ | $-7.98\%$ |
| FinGAN | 9.11% | 0.64 | 0.84 | 0.54 | $-16.99\%$ | $-1.76\%$ | $-8.16\%$ |
| Equal-weight | 9.10% | 0.64 | 0.83 | 0.55 | $-16.56\%$ | $-1.76\%$ | $-8.00\%$ |
| **Our model** | 8.98% | **0.65** | 0.83 | **0.60** | $\mathbf{-14.98\%}$ | $\mathbf{-1.73\%}$ | $\mathbf{-7.50\%}$ |

Our scenarios yield the best point estimates on every stress-aligned metric (MDD, Calmar, daily/36-day $\text{CVaR}_{95}$), with a non-uniform but diversified allocation (smallest weight 7.7%). A stationary-bootstrap check gives wide intervals, and the downside-risk advantage is consistent across resamples. Cross-asset dependence is corrected post-hoc rather than learned, which we leave to future work.

## E. Stylized-Fact Diagnostics

We evaluate seven standard stylized facts on the final model, aggregated over 4 seeds $\times$ 7 indices $\times$ 2 regimes ($= 56$ cells), against bootstrap CIs from real windows.

*Table E.1.* Stylized-fact pass rates for the final Dropout-FT model.

| Stylized fact | Pass rate | Verdict |
|---|---|---|
| OHLCV cross-channel consistency (Frobenius) | 56/56 | reproduced |
| Leverage effect, $\text{CCF}(r_t, r^2_{t+k>0}) < 0$ | 56/56 | reproduced |
| Non-Gaussianity (JB $p < 0.05$) | 55/56 | reproduced |
| Volatility clustering, $\text{ACF}(|r|)$ | 43/56 | largely present |
| Gain/loss asymmetry (skewness in real CI) | 32/56 | partial |
| Tail heaviness (Hill in real CI) | 25/56 | overshoots |
| No autocorrelation, $\text{ACF}(r)$ | 1/56 | not satisfied |

The generator reproduces the core structural facts strongly — leverage effect and OHLCV cross-channel consistency at 56/56, non-Gaussianity at 55/56 — and largely preserves volatility clustering (43/56). The two softer facts are consistent with a stress-oriented generator: mild tail overshoot (Hill) is expected and even desirable when the goal is to amplify extreme moves, and the kurtosis-aware fine-tuning that corrects tail ordering (Table B.3) trades a small amount of volatility-clustering fidelity for it. The one clear limitation is residual return-level autocorrelation (ACF(r), 1/56); we report it openly as a target for future work.

