# OpenReview forum: "Behavioral Proxy Conditioning for Financial Stress Scenario Generation with a Pretrained Diffusion Model"
_ICML.cc/2026/Workshop/FMSD — FMSD @ ICML 2026 Poster_

### Official Review · Reviewer_Ahxa · 2026-05-19
**A promising financial diffusion model but with insufficiently empirical evaluations**

**Rating:** 5
**Confidence:** 2

**Review:**

Summary: The paper proposes a conditional financial time series generation method for stress-scenario synthesis using a pretrained diffusion backbone, ImagenFew, adapted to seven global equity indices. The core idea is to condition generation on a five-dimensional behavioral proxy consisting of HMM-derived calm/normal/stress regime probabilities, cross-market correlation, and realized-volatility rank, with a market identity token. The paper further adds proxy-group dropout to reduce reliance on any single proxy component and a kurtosis-aware fine-tuning stage to improve stress-tail behavior. The experiment compares calm and stress generated windows against several baselines.

Strong Points:

1. Financial stress-scenario generation under crisis-data scarcity is an interesting time series problem, and the paper clearly motivates why purely historical replay and unconstrained generation are insufficient for stress testing in Section 2.

2. Adapting ImagenFew to financial time series is a sensible direction for data-scarce settings, and the inclusion of multiple ablation studies strengthens the empirical analysis.

Potential Weaknesses:

1. The paper includes some useful baselines but they are not sufficient for a convincing comparison in conditional generation. The paper could include more conditional generation models from GAN, VAE, and diffusion families.

2. Although the data are chronologically split, the reported stress protocol appears largely based on generated calm or stress samples conditioned by training-derived proxies. The paper does not clearly compare generated stress windows against held-out historical crisis periods, nor does it show that generated stress scenarios match unseen real stress distributions rather than just amplify volatility.

3. Most metrics used in Section 3 focus on generation fidelity, close-channel volatility, and tail losses. The paper uses OHLCV data, but it does not verify OHLC consistency constraints, autocorrelation, cross-asset dependence, or stylized facts beyond kurtosis.

---

### Official Review · Reviewer_2pRP · 2026-05-20
**Interesting proxy-conditioned diffusion approach, but evaluation is partly circular and scope is limited**

**Rating:** 5
**Confidence:** 3

**Review:**

**Summary**

This paper proposes a behavioral-proxy conditioning approach for financial stress scenario generation using a pretrained diffusion model. The model conditions generation on a compact five-dimensional proxy consisting of HMM regime probabilities, cross-market correlation, and realized volatility rank. The goal is to generate controllable calm and stress scenarios for seven global equity indices using daily OHLCV data from 2000–2024.

The final Dropout-FT model achieves strong reported regime separation, passing 27/28 stress-testing checks with 0% exact replay. The paper also includes ablations showing that proxy conditioning and LVN-off preprocessing are important for controllability.

Overall, the paper is clear, relevant, and practically motivated. However, the evaluation is not fully convincing because part of the stress/calm evaluation is closely tied to the conditioning signal itself, the empirical scope is narrow, and the comparison to strong conditional diffusion baselines is missing.

**Strengths**

The paper addresses an important and practical problem: generating financial stress scenarios when real crisis data is scarce.

The behavioral proxy is compact, interpretable, and financially meaningful. Combining HMM regime probabilities, cross-market correlation, and volatility rank is a reasonable way to encode market stress conditions.

The ablation study is useful. Removing the behavioral proxy substantially weakens stress/calm separation, and the LVN-on ablation supports the authors’ argument that local volatility normalization can remove important regime information.

The paper reports multi-seed stability and bootstrap confidence intervals, which improves confidence in the reported stress-separation results.


**Areas for Improvement**

The main concern is that the evaluation is partly circular. Stress and calm conditions are defined using volatility rank, and volatility rank is also part of the conditioning proxy. Therefore, showing that stress-conditioned samples have higher volatility is an important sanity check, but not strong independent evidence of realistic stress generation.

The empirical scope is limited. The model is evaluated only on seven equity indices, daily data, and 36-day windows. This is useful, but it is still a narrow setting for claims about financial stress scenario generation.

The contribution of pretraining is not isolated clearly enough. Since the method is built on a pretrained diffusion model, the paper should compare against a diffusion model trained from scratch or a randomly initialized version under the same conditioning setup.

The baseline comparison is incomplete. GBM, Historical Bootstrap, and FinGAN are useful references, but the most relevant comparisons would be other conditional diffusion models. The paper discusses CoFinDiff and InterDiff but does not include them as direct baselines.

The novelty analysis could be stronger. Exact replay is a strict criterion, since generated windows may be very close to historical windows without matching exactly. The nearest-neighbor analysis helps, but more detailed diversity and near-replay diagnostics would strengthen the claim.

The economic interpretation is somewhat approximate. The conversion back to raw-return units is described as approximate, yet stress testing is ultimately an economic/risk-management application. This part should be made more rigorous.

**Detailed Comments**

Please expand the evaluation beyond seven equity indices if possible. Additional asset classes such as FX, rates, commodities, or crypto would better support the stress-testing claim.

Please discuss whether generated stress scenarios correspond to economically plausible crisis patterns, not only higher volatility and worse CVaR. For example, do generated stress scenarios resemble known crisis periods in terms of drawdown shape, cross-market co-movement, and persistence?

Please make the raw-return conversion more rigorous. Since portfolio CVaR is central to the stress-testing motivation, approximate inverse normalization may not be sufficient for strong economic claims.


Please clarify the role of kurtosis-aware fine-tuning. Since the model is evaluated partly on kurtosis direction, an auxiliary kurtosis loss may make this metric less independent. This should be discussed more explicitly.


**Justification of Score**

I would assign this paper a 5: marginally below acceptance threshold.

The paper is clear, relevant, and technically reasonable. The behavioral proxy is interpretable, the ablations are useful, and the results show strong controllability under the proposed evaluation protocol. However, the empirical evidence is not fully convincing because the evaluation is partly tied to the conditioning signal, the scope is limited to seven daily equity-index series, the contribution of pretraining is not isolated, and strong conditional diffusion baselines are missing.

Overall, I view the paper as a promising preliminary contribution to controllable financial scenario generation, but it needs stronger independent validation and broader evaluation before the claims are fully convincing.